# Neurotrophin-3 regulates ribbon synapse density in the cochlea and induces synapse regeneration after acoustic trauma

Guoqiang Wan[1,2,3], Maria E Gómez-Casati[1,2], Angelica R Gigliello[1], M Charles Liberman[4,5], Gabriel Corfas[1,2,3,4]*†

[1]F M Kirby Neurobiology Center, Boston Children's Hospital, Boston, United States; [2]Department of Neurology, Harvard Medical School, Boston, United States; [3]Kresge Hearing Research Institute, Department of Otolaryngology - Head and Neck Surgery, University of Michigan, Ann Arbor, United States; [4]Department of Otology and Laryngology, Harvard Medical School, Boston, United States; [5]Eaton Peabody Laboratories, Massachusetts Eye and Ear Infirmary, Boston, United States

**\*For correspondence:** corfas@med.umich.edu

**Present address:** †Kresge Hearing Research Institute, Department of Otolaryngology - Head and Neck Surgery, University of Michigan, Ann Arbor, United States

**Competing interests:** The authors declare that no competing interests exist.

**Abstract** Neurotrophin-3 (Ntf3) and brain derived neurotrophic factor (Bdnf) are critical for sensory neuron survival and establishment of neuronal projections to sensory epithelia in the embryonic inner ear, but their postnatal functions remain poorly understood. Using cell-specific inducible gene recombination in mice we found that, in the postnatal inner ear, Bbnf and Ntf3 are required for the formation and maintenance of hair cell ribbon synapses in the vestibular and cochlear epithelia, respectively. We also show that supporting cells in these epithelia are the key endogenous source of the neurotrophins. Using a new hair cell CreER[T] line with mosaic expression, we also found that Ntf3's effect on cochlear synaptogenesis is highly localized. Moreover, supporting cell-derived Ntf3, but not Bbnf, promoted recovery of cochlear function and ribbon synapse regeneration after acoustic trauma. These results indicate that glial-derived neurotrophins play critical roles in inner ear synapse density and synaptic regeneration after injury.

## Introduction

The trophic factors, neurotrophin-3 (Ntf3) and brain-derived neurotrophic factor (Bbnf) play critical roles in the embryonic inner ear, contributing to the survival of cochlear and vestibular sensory neurons and to the establishment of their projections into the respective sensory epithelia (*Fritzsch et al., 2004*; *Ramekers et al., 2012*). Both Bdnf and Ntf3 continue to be expressed in inner ear sensory epithelia after birth. In the postnatal vestibular epithelia, Bdnf is expressed only by supporting cells (*Schecterson and Bothwell, 1994*; *Montcouquiol et al., 1998*), whereas Ntf3 is expressed by both supporting cells and hair cells (*Farinas et al., 1994*). In the postnatal organ of Corti, Bdnf is expressed by both inner (IHCs) and outer hair cells (OHCs) and supporting cells at early postnatal ages (P1–P6), but is only detected in supporting cells from P10 onwards (*Wiechers et al., 1999*). All cells in the early postnatal organ of Corti appear to express Ntf3, but in the adult, expression is restricted to the IHCs and their surrounding supporting cells, with higher levels in the apical (low-frequency) region than at the base of the cochlear spiral (*Sugawara et al., 2007*). Thus Bdnf and Ntf3 could have significant functions in the postnatal inner ear, and alterations in expression of these neurotrophins may modulate structure and function in the adult inner ear.

In this study, we investigated the roles of postnatal Ntf3 and Bdnf in both normal and damaged inner ears. Using cell-specific and inducible knockout or overexpression technology, we eliminated or increased neurotrophin expression from supporting cells or hair cells in the postnatal inner ear. We

**eLife digest** Noise-induced hearing loss is common, and can result from prolonged exposure to moderate levels of noise that are not perceived as painful or even unpleasant. Some hearing loss can be attributed to the death of hair cells in a part of the inner ear called the cochlea. When sound waves hit the cochlea, they cause the fluid inside it to vibrate: the hair cells detect these vibrations and convert them into electrical signals that are sent along neurons to the brain. However, vibrations that are too strong can destroy hair cells.

Increasing evidence suggests that hearing loss also results from damage to the synapses that connect the hair cells and the neurons in the cochlea. During development of the inner ear, molecules called growth factors are needed to ensure the survival of these neurons. Wan et al. predicted that these growth factors might also have a role in adult animals, and that producing more of them might help to safeguard hearing from the damaging effects of noise.

Consistent with this, mice that were genetically modified to lack a growth factor called neurotrophin-3 had cochleae that did not work properly and had fewer synapses between hair cells and neurons compared to control mice. Conversely, mice that produced too much neurotrophin-3 had more synapses than controls and also recovered more quickly from the effects of 2 hr exposure to 100 dB noise (roughly the volume of a pneumatic drill). Studies of the cochlea revealed that the extra neurotrophin-3 had boosted the regeneration of synapses damaged by the noise.

The beneficial effects of neurotrophin-3 were still seen when overproduction was started shortly *after* noise exposure, suggesting that it could have therapeutic potential. This is particularly significant in the light of recent evidence that the loss of synapses often comes before the death of hair cells in both age-related hearing loss and noise-induced hearing loss.

show that these neurotrophins, when expressed by the glia-like supporting cells in these sensory epithelia, are required for the formation and/or maintenance of ribbon synapses, a role distinct from the one they play in embryogenesis. Ntf3 has major effects only in the cochlea, while postnatal Bdnf appears to act only in the vestibular organs. Furthermore, we show that Ntf3 overexpression can elicit regeneration of the synaptic contacts between cochlear nerve terminals and inner hair cells after noise-induced synaptopathy, a type of neural damage which appears to be widespread even in ears exposed at sound levels well below those which cause hair cell damage and permanent threshold shifts (*Kujawa and Liberman, 2009*).

## Results

### Ntf3, but not Bdnf, expression by postnatal supporting cells regulates cochlear function

To alter neurotrophin expression by supporting cells, we used a mouse line that expresses the tamoxifen-inducible Cre recombinase (CreER$^T$) under the control of the proteolipid protein 1 promoter (*Plp1/CreER$^T$*). We previously showed that this line can be used to induce effective gene recombination in inner ear supporting cells, when tamoxifen is delivered at early postnatal ages (*Gomez-Casati et al., 2010a*). We crossed *Plp1/CreER$^T$* with mice carrying a conditional Ntf3 knockout allele (*Ntf3$^{flox}$*) (*Bates et al., 1999*) or an overexpression transgene (*Ntf3$^{stop}$*) to eliminate or increase the expression of Ntf3, respectively.

RT-qPCR showed that Ntf3 expression was significantly reduced in both the cochlea and utricle of *Ntf3$^{flox}$:Plp1/CreER$^T$* mice and increased in those of *Ntf3$^{stop}$:Plp1/CreER$^T$* mice after tamoxifen treatment (*Figure 1A*). Mice with either Ntf3 knockout (*Ntf3$^{flox}$:Plp1/CreER$^T$*) or overexpression (*Ntf3$^{stop}$:Plp1/CreER$^T$*) in supporting cells had normal vestibular evoked potentials (VsEPs, *Figure 1B,C*), indicating that postnatally, supporting cell-derived Ntf3 is not necessary for vestibular function. This is not the case for the embryonic inner ear, where the loss of Ntf3 results in the loss of a subpopulation of vestibular neurons (*Ernfors et al., 1995*). In contrast, *Ntf3* knockout from supporting cells resulted in elevated thresholds for auditory brainstem responses (ABRs) at high frequencies, while supporting cell Ntf3 overexpression resulted in reduced ABR thresholds at the same frequencies (*Figure 1E*). Distortion product otoacoustic emissions (DPOAEs) were not affected by the abnormal Ntf3 expression (*Figure 1D*),

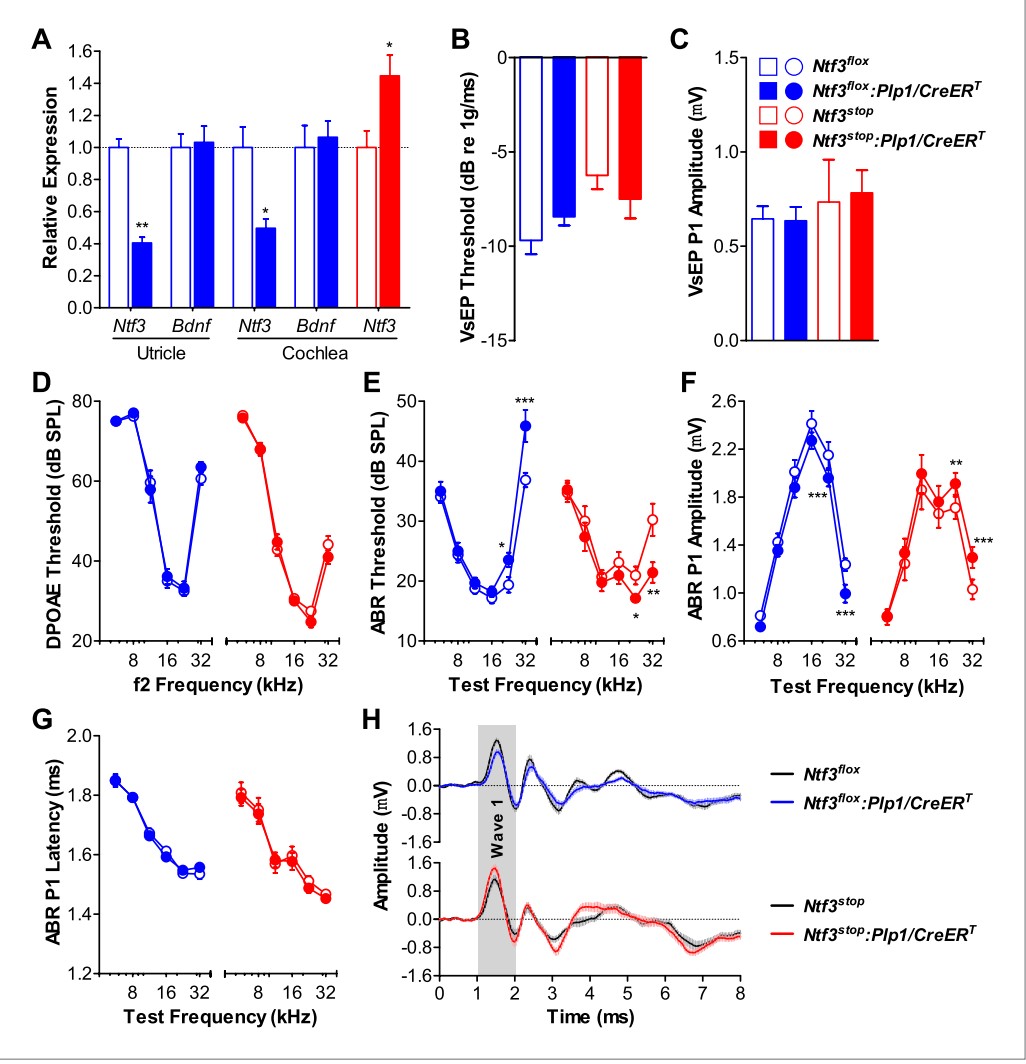

**Figure 1**. Ntf3 expression by postnatal supporting cells is required for cochlear, but not vestibular function. (**A**) RT-qPCR shows that postnatal tamoxifen injection reduces or increases Ntf3 mRNA levels in adult *Ntf3flox:Plp1/CreERT* or *Ntf3stop:Plp1/CreERT* inner ears, respectively; n = 5–6. *p < 0.05, **p < 0.01 by two-tailed unpaired t test. (**B** and **C**) Postnatal *Ntf3* knockout (blue) or overexpression (red) from supporting cells does not alter VsEP thresholds (**B**) or their peak 1 (P1) amplitudes at 0 dB (**C**); n = 4–8. (**D**–**F**) Postnatal knockout or overexpression of *Ntf3* from supporting cells reduces or enhances cochlear function, respectively. *Ntf3* knockout (blue) elevates ABR thresholds (**E**) and decreases ABR P1 amplitudes (**F**), without changing DPOAE thresholds (**D**); n = 16–17. *Ntf3* overexpression (red) reduces ABR thresholds (**E**) and increases ABR P1 amplitudes (**F**), without changing DPOAE thresholds (**D**); n = 21. ABR P1 amplitudes were assessed at 70 dB SPL. *p < 0.05, **p < 0.01, ***p < 0.001 by two-way ANOVA. (**G**) ABR P1 latencies are not affected by either *Ntf3* knockout (blue) or *Ntf3* overexpression (red) at all frequencies examined. Key in **C** applies to **A**–**G**. (**H**) Mean ABR waveforms from responses to 32 kHz tone pips from *Ntf3* knockouts and their controls (upper) and *Ntf3* overexpressors and their controls (lower). Gray shading indicates ABR wave 1. Both ABR P1 latencies (**G**) and waveforms (**H**) results were assessed at 70 dB SPL; n = 13–17.

The following figure supplement is available for figure 1:

**Figure supplement 1**. Expression of the *Plp1/CreERT* allele does not affect the cochlear function.

suggesting that the ABR changes were due to dysfunction of the inner hair cells (IHCs), the cochlear nerve fibers, or the synapses that connect them. Correspondingly, the amplitude of ABR peak 1 (P1), the summed activity of the cochlear nerve, was reduced by supporting cell *Ntf3* knockout and increased by overexpression, also at high frequencies (***Figure 1F***). The latency of ABR P1 and the width of ABR

wave 1 were not altered by the knockout or overexpression (*Figure 1G,H*), suggesting that altering Ntf3 levels does not affect cochlear nerve myelination. Control mice carrying only the *Plp1/CreER^T* allele showed normal cochlear responses, proving that the phenotype was not due to the expression of the *CreER^T* transgene (*Figure 1—figure supplement 1*). These results demonstrate that supporting cell derived Ntf3 plays a critical role in normal physiology of the postnatal cochlea, specifically in the basal (high-frequency) regions.

We then examined the role of supporting cell derived Bdnf in the postnatal inner ear using analogous conditional alleles (*Bdnf^flox* [*Rios et al., 2001*] and *Bdnf^stop* [*Chang et al., 2006*]) in combination with the *Plp1/CreER^T*. RT-qPCR showed that inner ear Bdnf expression was significantly altered (*Figure 2A*). Neonatal knockout of *Bdnf* in supporting cells resulted in elevated VsEP thresholds and decreased VsEP peak 1 (P1) amplitudes (*Figure 2B,C*; blue). This phenotype is identical to that seen when supporting cell *Bdnf* knockout was induced during late embryogenesis (E14.5-17.5) (*Gomez-Casati et al., 2010b*), demonstrating that the postnatal period is when supporting cell-derived Bdnf plays a critical role in the vestibular epithelium. Surprisingly, vestibular function was unaffected in mice overexpressing Bdnf (*Figure 2B,C*; red), indicating that Bdnf is expressed at sufficient amounts in the postnatal vestibular epithelium. Cochlear function was unaffected by either increased or reduced Bdnf expression as seen by DPOAE thresholds (*Figure 2D*), ABR thresholds (*Figure 2E*), ABR amplitudes (*Figure 2F*), and ABR latencies (*Figure 2G,H*), demonstrating that supporting cell-derived Bdnf is not important in the postnatal cochlea. Together, our results show that supporting cell-derived Ntf3 and Bdnf play complementary roles in the postnatal cochlea and vestibular organs.

## Supporting cell-derived Ntf3 regulates ribbon synapse density in postnatal cochlea

We then analyzed cochlear histopathology to determine the anatomical basis for the pathophysiology in *Ntf3* mutants. Examination of plastic sections showed no obvious changes in the organ of Corti, the spiral ganglion, or accessory structures of the cochlear duct such as the stria vascularis, spiral ligament, etc. Quantitative analysis showed normal density of supporting cells and hair cells (*Figure 3A,C*) as well as peripheral axons of cochlear nerve fibers (*Figure 3B,D*). In contrast, analysis of immunostained whole-mounts of the organ of Corti revealed alterations in the numbers of synaptic contacts between inner hair cells (IHCs) and cochlear nerve terminals. Immunostaining of pre-synaptic ribbon and post-synaptic receptor patches, using antibodies against CtBP2 and GluA2, respectively (*Figure 4A*), showed that supporting cell *Ntf3* knockout reduced the number of synaptic puncta (*Figure 4A*; upper panels and *Figure 4D*; blue), while overexpression increased synapse number (*Figure 4A*; lower panels and *Figure 4D*; red). Effects on synaptic counts were restricted to basal (high-frequency) cochlear regions (*Figure 4D*), as seen for ABR thresholds and amplitudes (*Figure 1E,F*). Similar results were obtained when separately counting either pre-synaptic ribbons or post-synaptic receptor patches (*Figure 4B,C*). The strong correlation between the changes in ABR amplitudes and synaptic counts (*Figure 4E*) suggests that levels of supporting cell-derived Ntf3 influence cochlear function by regulating the number of IHC ribbon synapses. Previous studies have suggested that alterations in ribbon synapses affect ABR responses by changing sound-evoked peak discharge rates (*Davies, 1996*; *Glowatzki et al., 2006*; *Gomez-Palacio-Schjetnan and Escobar, 2013*). Thus, increases (or decreases) in the number of synapses per fiber in *Ntf3* overexpressors or knockouts could increase (or decrease) the probability of synchronous EPSPs, thereby increasing (or decreasing) onset spike rate in response to tone pips and ultimately altering the amplitude of ABR P1, the summed activity of the cochlear nerve. The observation that Ntf3 overexpression can increase synapse density indicates that Ntf3 is normally expressed in limited amounts, as expected from the neurotrophic hypothesis (*Davies, 1996*).

## Roles of hair cell-derived Ntf3 in postnatal cochlea

Since both supporting cells and IHCs express Ntf3 in the postnatal organ of Corti (*Sugawara et al., 2007*), it was important to assess the contribution of hair cell-derived Ntf3. We generated a mouse line carrying tamoxifen-inducible Cre under the control of the *Pou4f3* promoter (*Pou4f3/CreER^T*), which is active only in hair cells in the postnatal inner ear (*Fujioka et al., 2011*). Using a *Rosa26^tdTomato* reporter line (*Madisen et al., 2010*), we found that the *Pou4f3/CreER^T* line drives inducible gene recombination in about 50% of inner and outer hair cells, with similar recombination efficiency at all cochlear regions (*Figure 5A*), and with a pattern similar to another hair cell promoter, *Atoh1/CreER^T* (*Weber et al., 2008*). We generated mice with knockout or overexpression of *Ntf3* in hair cells by crossing

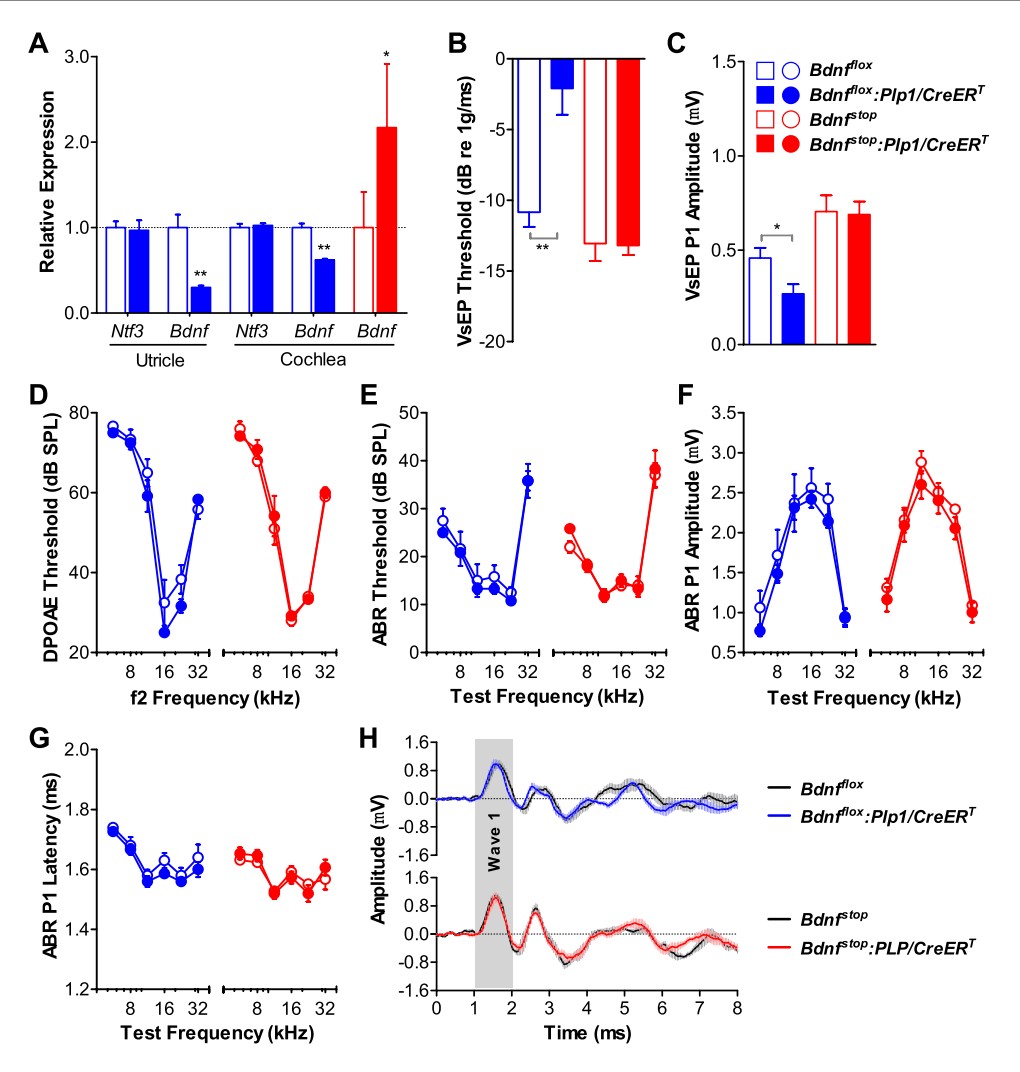

**Figure 2.** Bdnf expression by postnatal supporting cells is required for vestibular, but not cochlear function. (**A**) RT-qPCR shows that postnatal tamoxifen injection results is reduced or increased Bdnf mRNA levels in adult *Bdnf^flox^:Plp1/CreER^T^* or *Bdnf^stop^:Plp1/CreER^T^* inner ears, respectively; n = 6. (**B** and **C**) Postnatal *Bdnf* knockout (blue) from supporting cells leads to higher VsEP thresholds (**B**) and lower VsEP P1 amplitudes (**C**) at the high stimulus level (0 dB *re* 1 g/ms). Postnatal *Bdnf* overexpression (red) does not affect the vestibular function (**B** and **C**); n = 6–11. *p < 0.05; **p < 0.01 by two-tailed unpaired t tests. (**D–H**) Neither *Bdnf* knockout (blue) nor overexpression (red) from postnatal supporting cells alters cochlear function, as shown by normal DPOAE thresholds (**D**), ABR thresholds (**E**), ABR P1 amplitudes (**F**), ABR P1 latencies (**G**), and ABR P1 waveforms (**H**) at 70 dB SPL compared to control littermates; n = 6. Key in **C** applies to **A–G**. Gray shading (**H**) indicates ABR wave 1.

*Pou4f3/CreER^T^* with *Ntf3^flox^* or *Ntf3^stop^* mice and treating them with tamoxifen at early postnatal ages. RT-qPCR analysis showed that Ntf3 expression was significantly reduced and increased in cochleae of *Ntf3* knockouts and overexpressors, respectively (**Figure 5B**). Mice with hair cell-specific *Ntf3* knockout had no cochlear phenotype: ABR thresholds and P1 amplitudes were indistinguishable from controls (**Figure 5C,D**; blue). Thus, hair cell-derived Ntf3 appears to be dispensable in the postnatal cochlea. In contrast, mice overexpressing Ntf3 in postnatal hair cells showed significant reduction in ABR threshold at 32 kHz (**Figure 5C**; red) as well as increased ABR P1 amplitudes at frequencies ≥16 kHz (**Figure 5D**; red), similar to the patterns seen for supporting cell specific Ntf3 overexpression (**Figure 1**).

To probe the mechanisms underlying the effects of hair cell-derived Ntf3 overexpression, we analyzed IHC synaptic counts. For this, we generated *Ntf3^stop^:Rosa26^tdTomato^:Pou4f3/CreER^T^* mice, which

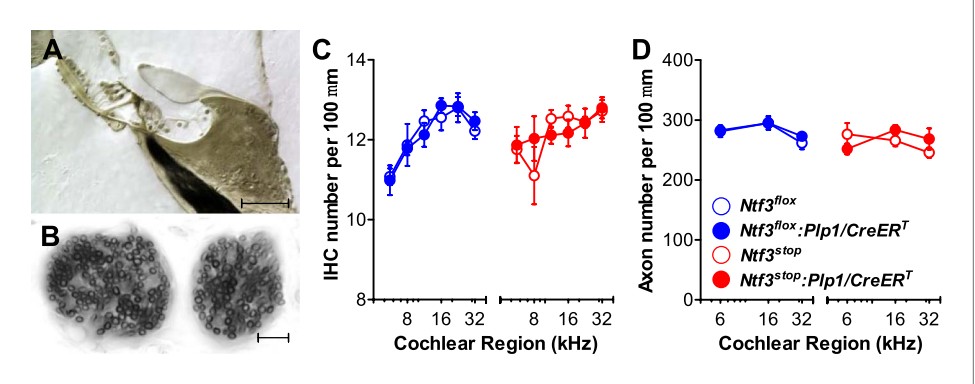

**Figure 3**. Ntf3 expression by postnatal supporting cells does not affect the organ of Corti morphology, hair cell or axon numbers. (**A** and **B**) Photomicrographs of the basal turn of an *Ntf3^{flox}:Plp1/CreER^T* cochlea (~32 kHz region) showing a cross-section of the organ of Corti (**A**; scale bar = 100 μm) and a tangential section through the osseous spiral lamina showing peripheral axons of cochlear nerve fibers (**B**; scale bar = 10 μm). (**C**) The number of IHCs per 100 μm of organ of Corti is not altered by either *Ntf3* knockout (blue) or overexpression (red) at any cochlear regions; *n* = 5–6. (**D**) The number of cochlear-nerve peripheral axons per 100 μm of osseous spiral lamina (near the habenula perforata) is not affected by either *Ntf3* knockout (blue) or overexpression (red); *n* = 6. Key in **D** applies to **C** and **D**.

allowed identification of hair cells in which the *Ntf3* overexpression transgene had been activated, based on tdTomato fluorescence (*Figure 5E*). Pre-synaptic ribbons and post-synaptic receptor patches were increased specifically in the recombined (tdTomato (+) and *Ntf3* overexpressing) IHCs (*Figure 5F*). This observation suggests that the Ntf3 released by IHCs remains close to its source, possibly because the supporting cells create a diffusion barrier, preventing cross talk between adjacent IHCs. Together, these results show that supporting cells are the key endogenous source of Ntf3 for synaptic survival, but that Ntf3 overexpression in hair cells can increase synaptic density, even in the postnatal cochlea.

### Overexpression of Ntf3, but not Bdnf, by postnatal supporting cells promotes recovery of cochlear responses after acoustic trauma

A significant component of the acute response to acoustic trauma (AT) is the swelling of cochlear nerve terminals at afferent synapses, suggestive of noise-induced glutamate excitotoxicity (*Robertson, 1983*). Such damage can lead to a permanent loss of IHC synapses and reduction in ABR amplitudes, followed by a slow death of spiral ganglion neurons, despite complete recovery of cochlear thresholds within 1–2 weeks (*Kujawa and Liberman, 2009*). To determine if Ntf3 overexpression could either prevent, or promote recovery from, this noise-induced synaptic loss and attenuation of cochlear responses, we exposed control and *Ntf3* overexpressing mice to noise levels known to cause this type of neuropathy (8–16 kHz at 100 dB SPL for 2 hr) (*Kujawa and Liberman, 2009*). In *Ntf3^{stop}* control mice, this exposure caused immediate threshold shifts as large as 50 dB, which recovered partially by 3 days and almost completely (except 32 kHz) by 10 days after exposure (*Figure 6A*; left panel). However, even after 10 days, ABR P1 amplitudes (at ≥16 kHz) remained significantly lower than before exposure (*Figure 6B*; left panel). In contrast, in *Ntf3* overexpressing mice, ABR thresholds recovered completely by 3 days post-exposure (*Figure 6A*; right panel and *Table 1*), and P1 amplitude recovery was significantly enhanced (*Figure 6B*; right panel and *Table 1*). In contrast to Ntf3, a parallel set of experiments on mice overexpressing Bdnf in supporting cells showed no effects on the extent or time course of the noise-induced pathophysiology (*Figure 6C,D* and *Table 1*).

### Overexpression of Ntf3 by postnatal supporting cells promotes synaptic regeneration after acoustic trauma

Analysis of immunostained cochlear epithelia from the noise-exposed ears showed that Ntf3 overexpression does not prevent noise-induced synaptic loss, but rather supports synapse regeneration. Immediately after exposure (AT + 2 hr), control and *Ntf3* overexpressing mice had the same degree of synaptic loss (colocalized GluA2–CtBP2 puncta, *Figure 7A*; upper panels, *Figure 7B* and *Table 2*). Similarly, Ntf3 overexpression did not alter the acute loss of pre-synaptic ribbons (*Figure 7C* and

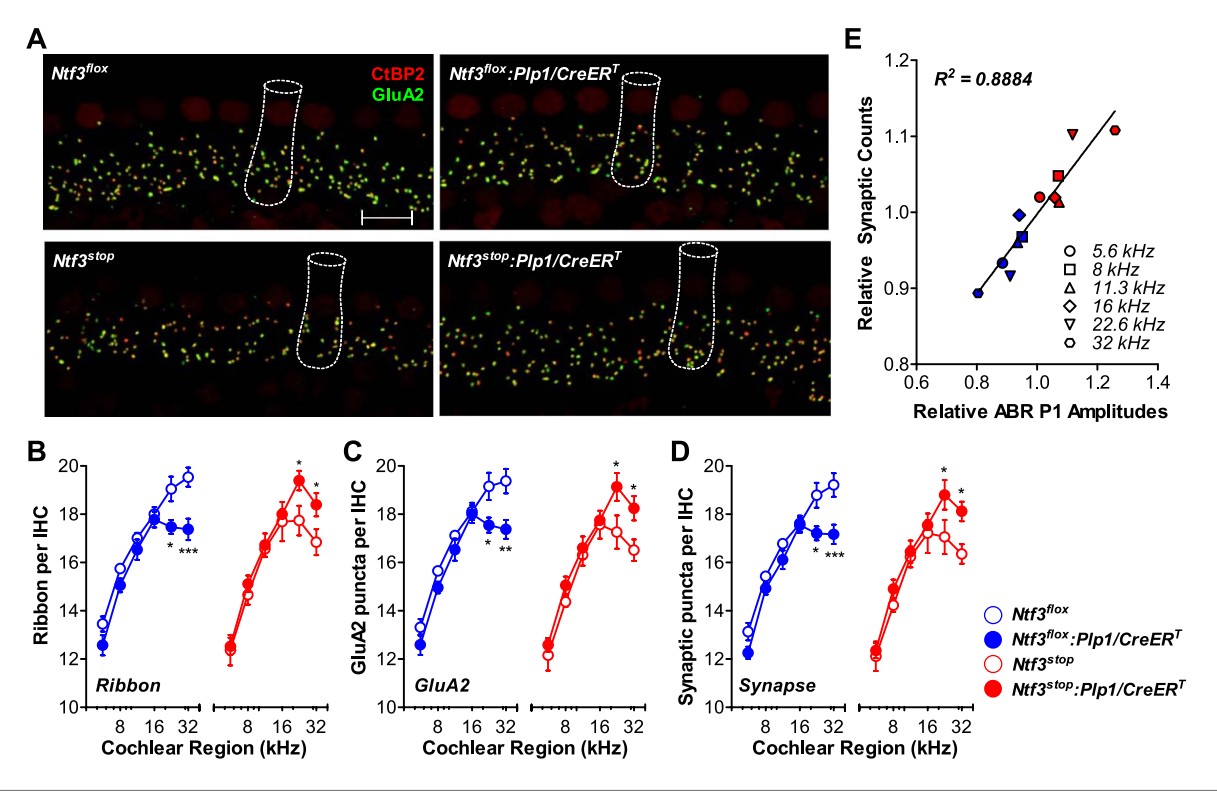

**Figure 4**. Ntf3 expression by postnatal supporting cells regulates hair cell ribbon synapse density at high frequencies. (**A**) Representative confocal images (maximal projection from a focal series) of IHC synapses from 32 kHz region of *Ntf3^flox^*, *Ntf3^flox^:Plp1/CreER^T^*, *Ntf3^stop^*, and *Ntf3^stop^:Plp1/CreER^T^* cochleae immunolabeled for pre-synaptic ribbons (CtBP2-red) and post-synaptic receptor patches (GluA2-green) (scale bar = 10 µm). The dashed lines show the approximate outline of one IHC. CtBP2 antibody also weakly stains IHC nuclei. (**B–D**) Quantitative data shows that *Ntf3* knockout reduces, and overexpression increases, the number of pre-synaptic ribbons (**B**), post-synaptic GluA2 receptor patches (**C**), and putative ribbon synapses, defined as juxtaposed CtBP2- and GluA2-positive puncta (**D**) at high frequency cochlear regions; n = 5–6. *p < 0.05, **p < 0.01, ***p < 0.001 by two-way ANOVA. (**E**) Relative synaptic counts vs relative ABR P1 amplitudes of *Ntf3* knockouts (blue) or overexpressors (red) shows a linear correlation. Data points were obtained by normalizing synaptic counts and ABR P1 amplitudes of *Ntf3* mutants to the values of their respective controls at each of the frequency regions analyzed. Key in **D** applies **B–E**.

*Table 2*). However, at 14 days of post-exposure (AT + 14 days), a clear effect of Ntf3 on IHC synapses was observed. Whereas the synaptic loss in control mice remained equal to that in AT + 2 hr, *Ntf3* overexpressors showed significant synaptic recovery (*Figure 7A*; lower panels, *Figure 7B* and *Table 2*). In control mice, noise-induced loss of pre-synaptic ribbons was progressive, with further reductions seen between 2 hr and 14 days post-exposure. Remarkably, Ntf3 overexpression prevented the progressive loss of synaptic ribbons (*Figure 7C* and *Table 2*).

## Overexpression of Ntf3 by adult supporting cells after acoustic trauma promotes recovery of auditory response and regeneration of ribbon synapses

For Ntf3 to be a viable candidate to treat noise-induced hearing loss, it must be effective even if applied after noise exposure. To test if Ntf3 overexpression after noise exposure enhances synaptic regeneration and functional recovery, we could not use the *Plp1/CreER^T^* transgene, as this line does not produce effective recombination in cochlear supporting cells after P15 (data not shown). Therefore, we used a line expressing *CreER^T^* under the control of the promoter for the glutamate aspartate transporter *Slc1a3* (*Slc1a3/CreER^T^*) (*Wang et al., 2012*). In the adult cochlea, Slc1a3 (GLAST) is expressed by inner phalangeal and inner border cells (*Furness and Lawton, 2003*; *Glowatzki et al., 2006*), the same supporting cells that express Plp1 at early postnatal ages (*Gomez-Casati et al., 2010a*). RT-qPCR showed that tamoxifen treatment in adult *Ntf3^stop^:Slc1a3/CreER^T^* mice significantly increased cochlear

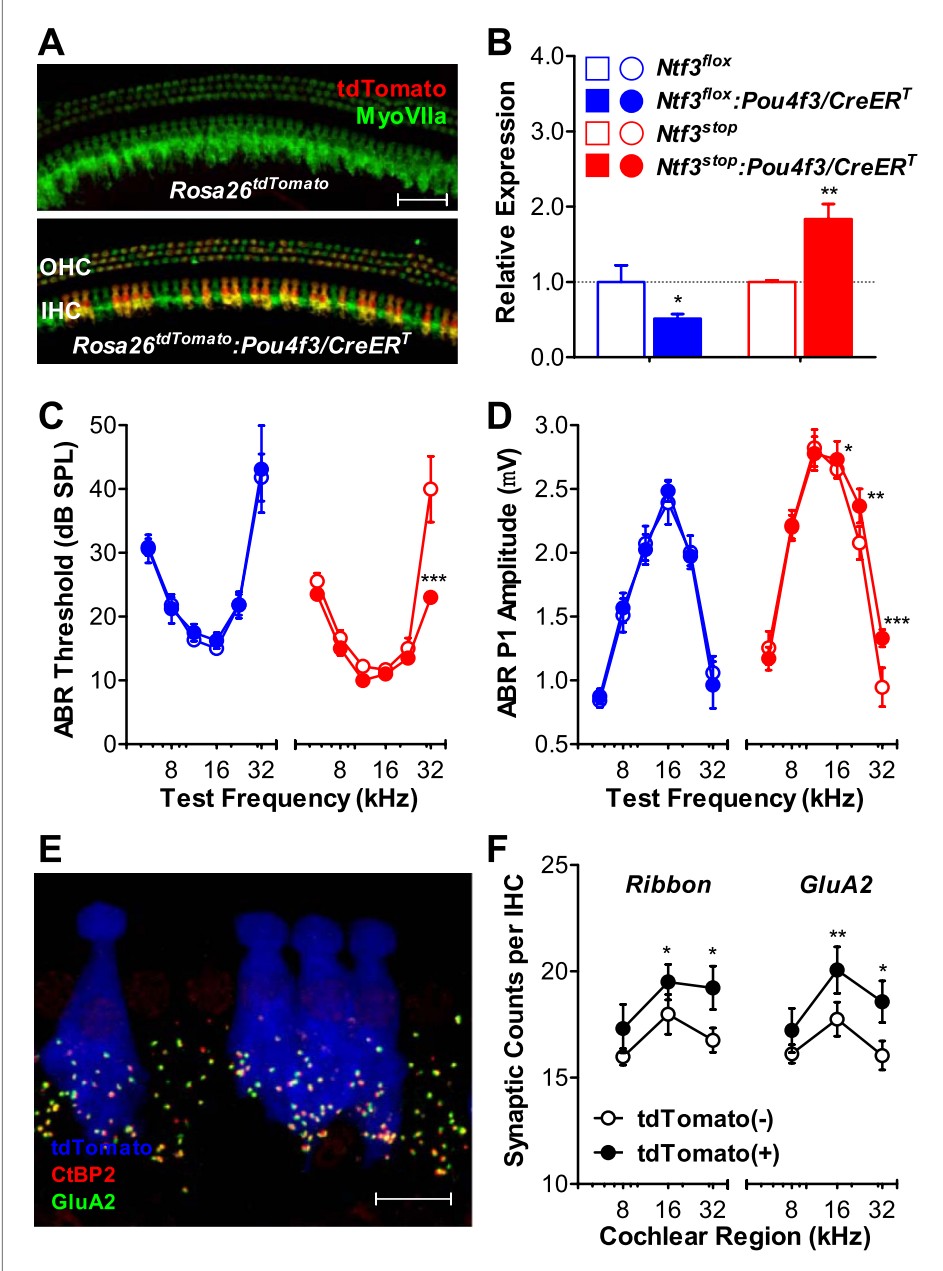

**Figure 5**. Ntf3 overexpression by postnatal hair cells increases cochlear sensitivity and synaptic densities at high frequencies. (**A**) *Pou4f3/CreER^T* allows for hair cells specific inducible gene recombination. *Rosa26^tdTomato:Pou4f3/CreER^T* mice and their littermate *Rosa26^tdTomato* controls were injected with tamoxifen at P1-P3, and the cochleas were collected at P60. Inducible recombination is seen as tdTomato fluorescence co-localized with MyoVIIa immunostaining. Scale bar = 50 μm. (**B**) RT-qPCR shows that postnatal tamoxifen injection reduced Ntf3 mRNA in *Ntf3^flox:Pou4f3/CreER^T* cochlea and increased Ntf3 expression in *Ntf3^stop:Pou4f3/CreER^T* cochlea, compared to their respective controls; n = 4–5. *p < 0.05, **p < 0.01 by two-tailed unpaired t tests. (**C** and **D**) Postnatal overexpression of *Ntf3* in hair cells (red) reduced ABR thresholds (**C**) and increased ABR P1 amplitudes (**D**) at high frequencies; n = 9–10. Postnatal knockout of *Ntf3* from hair cells (blue) had no effect on these measures; n = 8–11. ABR P1 amplitudes were assessed at 70 dB SPL. *p < 0.05, **p < 0.01, ***p < 0.001 by two-way ANOVA. (**E**) Confocal maximal projection of 7 adjacent IHCs from the 32 kHz region of a hair cell-specific *Ntf3* overexpressor, after immunostaining for synapses as in *Figure 4A*; tdTomato indicates recombined hair cells. Scale bar = 10 μm. (**F**) Synaptic counts are increased in recombined (tdTomato+, *Ntf3* overexpressing) IHCs compared to neighboring unrecombined cells; n = 6 cochleae, with at least 100 hair cells in each group. *p < 0.05, **p < 0.01 by two-tailed paired t tests. Key in **B** applies to **B–D**.

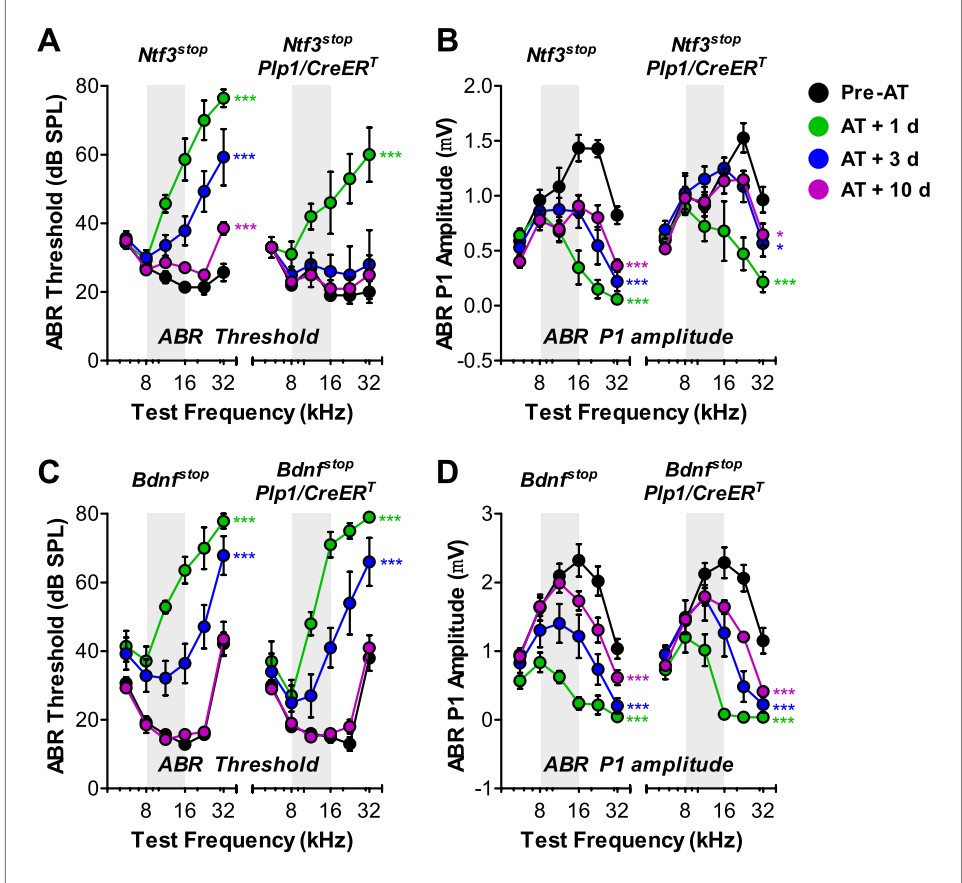

**Figure 6**. Overexpression of Ntf3, but not Bdnf, promotes recovery from noise-induced attenuation of cochlear responses. (**A** and **B**) *Ntf3* overexpression accelerates the recovery of ABR thresholds (**A**) and promotes recovery of ABR P1 amplitudes (**B**) after acoustic trauma (AT); n = 5–7. (**C** and **D**) *Bdnf* overexpression does not affect the recovery of ABR thresholds (**A**) and ABR P1 amplitudes (**B**) after acoustic trauma; n = 5–7. *p < 0.05, ***p < 0.001 by two-way ANOVA. Gray shading indicates the noise exposure spectrum. Key in **B** applies to all panels.

Ntf3 expression as compared to controls lacking the Cre (*Figure 8A*). *Ntf3^stop^:Slc1a3/CreER^T^* mice and control mice were then subjected to acoustic trauma, followed immediately (<1 hr) by tamoxifen treatment (AT/Tmx; *Figure 8B*). As for mice injected with tamoxifen in the early postnatal period (*Figure 6A*), controls and *Ntf3* overexpressors showed comparable loss and gradual recovery of ABR thresholds

**Table 1.** Statistical analysis (two-way ANOVA) of ABR threshold and P1 amplitude changes between control and neurotrophin overexpressing (*Plp1/CreER^T^*) mice after acoustic trauma

|  |  | *Ntf3* overexpressor vs control | | *Bdnf* overexpressor vs control | |
| --- | --- | --- | --- | --- | --- |
|  |  | p Value | F Statistic | p Value | F Statistic |
| ABR threshold | AT + 1 day | 0.0051 | 8.469 | 0.6801 | 0.1716 |
|  | AT + 3 days | <0.0001 | 19.00 | 0.6752 | 0.1774 |
|  | AT + 10 days | <0.0001 | 17.34 | 0.9862 | 0.0003 |
| ABR P1 amplitude | AT + 1 day | 0.0342 | 4.695 | 0.1899 | 1.7580 |
|  | AT + 3 days | <0.0001 | 18.08 | 0.5692 | 0.3276 |
|  | AT + 10 days | <0.0001 | 18.13 | 0.0421 | 4.3160 |

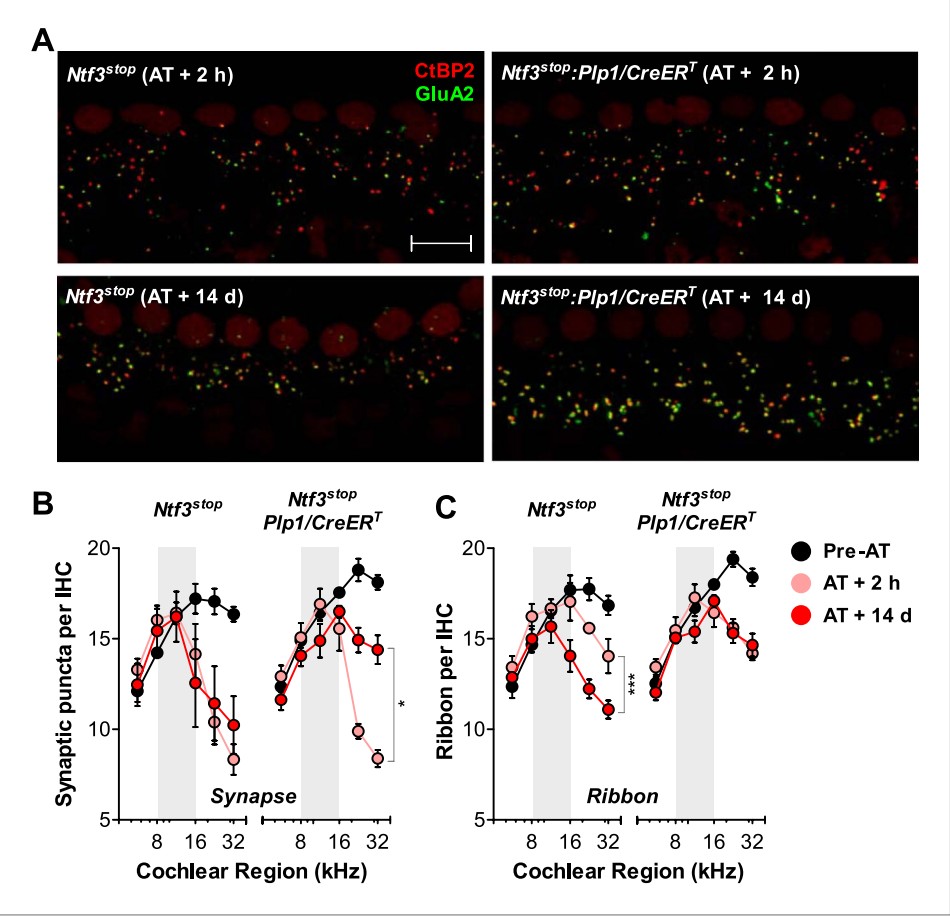

**Figure 7**. Ntf3 overexpression from postnatal supporting cells promotes recovery from noise-induced synaptic degeneration. (**A**) Representative confocal images of IHC synapses from 32 kHz region of *Ntf3stop* and *Ntf3stop:Plp1/CreERT* cochleae immunolabeled for CtBP2 and GluA2 at 2 hr (upper panels) or 14 days after acoustic trauma (AT); scale bar = 10 μm. (**B** and **C**) *Ntf3* overexpression promotes regeneration of IHC synapses (**B**) and prevents the progressive loss of IHC ribbons (**C**) between 2 hr (AT + 2 hr) and 14 days (AT + 14 days) after acoustic trauma. *n* = 3–8. *p < 0.05, ***p < 0.001 by two-way ANOVA. Gray shading indicates the noise exposure spectrum. Key in **C** applies to **B**–**C**.

when tamoxifen was administered in adult mice immediately after noise exposure (*Figure 8C* and *Table 3*). In contrast, the recovery of ABR P1 amplitudes was significantly enhanced by Ntf3 overexpression (*Figure 8D*). Interestingly, Ntf3 had no significant effects on ABR P1 amplitudes 3 days after noise, but by 14 days after exposure, *Ntf3* overexpressors showed significantly higher ABR P1 amplitudes compared to control littermates (*Figure 8D* and *Table 3*). The delay in Ntf3's impact on cochlear responses might suggest a slow increase in Ntf3 availability or a delay in the effects of Ntf3 on synapse regeneration.

To define the cellular mechanisms of the Ntf3 effects, we analyzed the density of ribbon synapses before and after noise exposure in these animals. As expected, the basal level of synaptic counts was similar between *Cre⁻* and *Cre⁺* mice (*Figure 8E*, top panels and *Figure 8F*). Importantly, 14 days after noise exposure and tamoxifen

**Table 2.** Statistical analysis (two-way ANOVA) of synaptic density changes between control and *Ntf3* overexpressing (*Plp1/CreERT*) mice after acoustic trauma

|         |              | p Value | F Statistic |
|---------|--------------|---------|-------------|
| Synapse | AT + 2 hr    | 0.9732  | 0.001       |
|         | AT + 14 days | 0.0065  | 8.358       |
| Ribbon  | AT + 2 hr    | 0.8134  | 0.056       |
|         | AT + 14 days | 0.0002  | 15.54       |

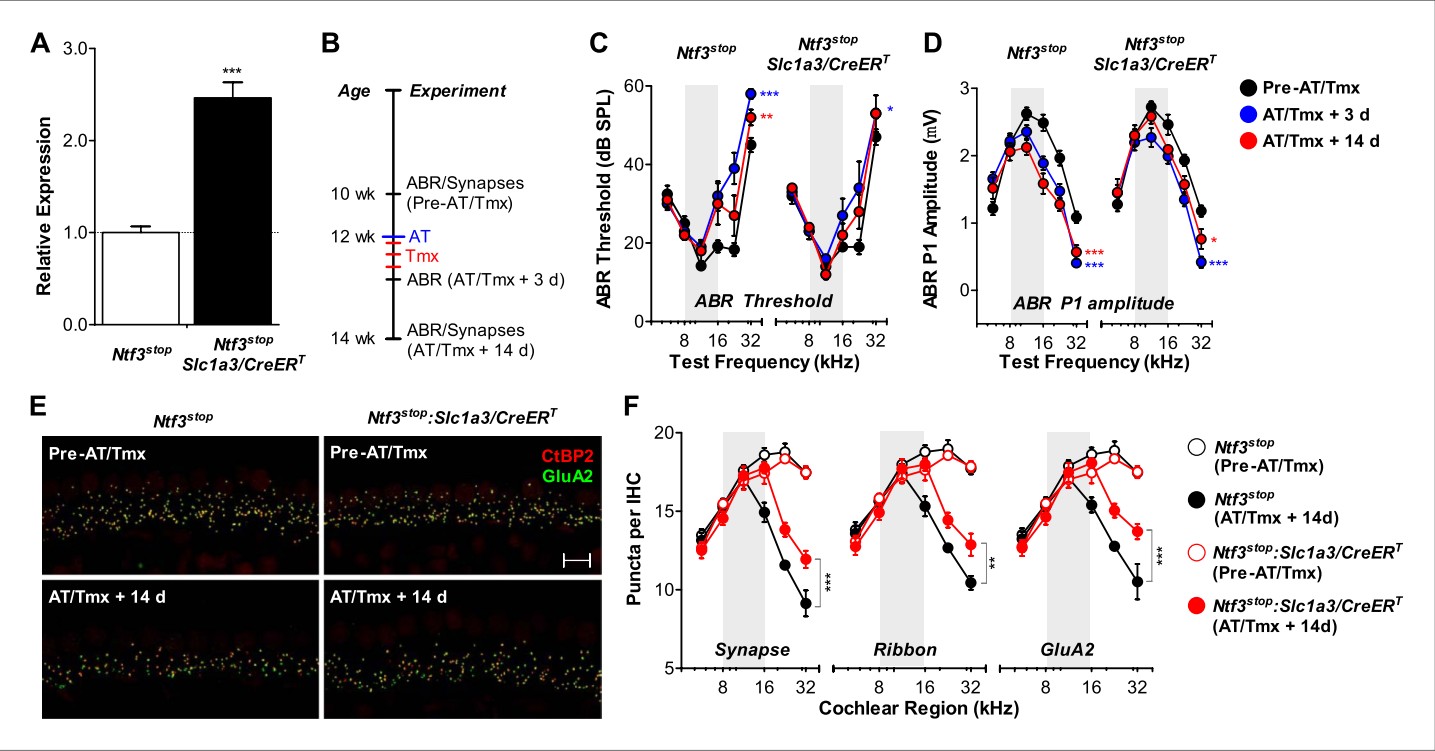

**Figure 8**. Ntf3 overexpression from adult supporting cells after acoustic trauma promotes auditory function recovery and synaptic regeneration. (**A**) RT-qPCR shows that tamoxifen treatments of adult mice increased Ntf3 expression in *Ntf3^stop^:Slc1a3/CreER^T^* cochlea; *n* = 6. \*\*\*p < 0.001 by two-tailed unpaired *t* tests. (**B**) Time line of the experiment showing the ages of mice for ABR measurements, acoustic trauma (AT), tamoxifen inductions (Tmx), and sample collections for synaptic counts. (**C–D**) The effects of *Ntf3* overexpression from adult supporting cells on ABR thresholds (**C**) and P1 amplitudes (**D**) after noise exposure; *n* = 5–9. \*p < 0.05, \*\*p < 0.01, \*\*\*p < 0.001 by two-way ANOVA. Key in **D** applies to **C–D**. (**E**) Representative confocal images of IHC synapses from 32 kHz region of *Ntf3^stop^* and *Ntf3^stop^:Slc1a3/CreER^T^* cochleae immunolabeled for CtBP2 and GluA2. The samples were collected from mice without AT/Tmx (Pre-AT/Tmx) or 14 days after AT/Tmx (AT/Tmx + 14 days). Scale bar = 10 μm. (**F**) *Ntf3* overexpression after acoustic trauma promotes regeneration of IHC ribbon synapses; *n* = 5–6. \*\*p < 0.01, \*\*\*p < 0.001 by two-way ANOVA. Gray shading indicates the noise exposure spectrum.

treatment, cochleae from *Ntf3* overexpressors had significantly higher density of pre-synaptic ribbons, post-synaptic GluA2 receptor patches, and putative synapses (*Figure 8E*, bottom panels and *Figure 8F*). Together with the results from *Ntf3^stop^:Plp1/CreER^T^* mice (*Figure 6*), our study demonstrates that Ntf3 overexpressed either before or immediately after noise exposure can effectively promote recovery from noise-induced cochlear synaptopathy.

**Table 3.** Statistical analysis (two-way ANOVA) of ABR threshold and P1 amplitude changes between control and *Ntf3* overexpressing (*Slc1a3/CreER^T^*) mice before and after acoustic trauma and tamoxifen treatments

| | | p Value | F Statistic |
|---|---|---|---|
| ABR threshold | Pre-AT/Tmx | 0.7774 | 0.081 |
| | AT/Tmx + 3 days | 0.1674 | 1.965 |
| | AT/Tmx + 14 days | 0.5301 | 0.400 |
| ABR P1 amplitude | Pre-AT/Tmx | 0.2826 | 1.166 |
| | AT/Tmx + 3 days | 0.3695 | 0.821 |
| | AT/Tmx + 14 days | 0.0008 | 12.71 |

## Discussion

Neurotrophins are key molecular mediators for synaptic development and function in the central nervous system (*Gomez-Palacio-Schjetnan and Escobar, 2013*). In this study, we show that Ntf3 and Bdnf, which are necessary for sensory neuron survival in the developing inner ear (*Fritzsch et al., 2004*; *Ramekers et al., 2012*), continue to play important and complementary roles in the postnatal inner ear, specifically by modulating the number of synapses between hair cells and sensory neurons. Our results indicate that supporting cells of the sensory epithelia are the key source of these neurotrophins in the postnatal inner ear. Importantly, we show that increasing

the availability of Ntf3, but not Bdnf, promotes the recovery of both cochlear responses and IHC synapses after acoustic trauma, even when Ntf3 expression is induced after noise exposure.

The specificity of the effects of each neurotrophin on each organ cannot be explained simply by the spatio-temporal expression pattern of the components for these signaling pathways, as both Bdnf and Ntf3, as well as their respective receptors Ntrk2 (TrkB) and Ntrk3 (TrkC), are expressed in the postnatal and adult cochlea and vestibular organs (*Pirvola et al., 1992*; *Fritzsch et al., 1999*; *Gestwa et al., 1999*; *Wiechers et al., 1999*; *Farinas et al., 2001*; *Stankovic and Corfas, 2003*; *Sugawara et al., 2007*). A similar specificity in the biological roles of these neurotrophins has been shown in embryonic development, that is, constitutive knockout of *Bdnf* or *Ntf3* reveals predominant pro-survival roles on vestibular or cochlear neurons, respectively (*Farinas et al., 1994*; *Jones et al., 1994*; *Ernfors et al., 1995*; *Schimmang et al., 1995*; *Bianchi et al., 1996*). However, while genetic replacement of one neurotrophin with the other can almost completely rescue neuronal survival deficits caused by constitutive deletion of either *Bdnf* or *Ntf3* (*Coppola et al., 2001*; *Agerman et al., 2003*), we observed that Ntf3 and Bdnf have distinct and non-overlapping roles in postnatal synapse formation. Thus, it appears that vestibular and auditory primary sensory neurons can respond equally to endogenous levels of either Bdnf or Ntf3 during development, but this ability is lost after birth.

In the vestibular system, supporting cell-derived Bdnf is the sole neurotrophin necessary for postnatal formation and maintenance of hair cell synapses (*Gomez-Casati et al., 2010b*), and the levels of Bdnf expressed by these non-neuronal cells is not limiting, as Bdnf overexpression does not alter vestibular function. While previous studies of constitutive knockouts showed that Ntf3 is necessary for the survival of a subpopulation of vestibular neurons during embryogenesis (*Ernfors et al., 1995*), our data indicate that Ntf3, which like Bdnf is expressed primarily by supporting cells in the postnatal vestibular sensory epithelia (*Sugawara et al., 2007*), is dispensable in the vestibular system after birth. In contrast to the vestibular organs, supporting cell-derived Ntf3, but not Bdnf, is necessary for the establishment of normal synapses and auditory function in the cochlear base. The correlation of Ntf3 expression levels with synapse numbers and cochlear sensitivity in both knockout and overexpression models indicates that supporting cell-derived Ntf3 is not only a critical but also a limiting factor in the postnatal cochlea.

Interestingly, postnatal knockout of supporting cell-derived Ntf3 and Bdnf did not affect the survival of the sensory neurons themselves, in either the vestibular (*Gomez-Casati et al., 2010b*) or the cochlear (spiral) ganglion, indicating that endogenous Ntf3 and Bdnf become dispensable for neuronal survival after birth. These sensory neurons may become independent of trophic support in the adult, or alternatively, other trophic factors, such as insulin-like growth factor-1 (Igf1) and macrophage migration inhibitory factor (Mif), may promote survival after birth. Both Igf1 and Mif are expressed in the postnatal cochlea, and, for both, loss results in ganglion cell death or altered innervation after birth (*Camarero et al., 2001*; *Bank et al., 2012*). In addition, glial cell line-derived neurotrophic factor (Gdnf) is also expressed in postnatal inner ear (*Stankovic and Corfas, 2003*), and it has been suggested that vestibular neurons switch trophic sensitivity from Bdnf to Gdnf after target innervation (*Hashino et al., 1999*).

Although adult inner hair cells also express Ntf3 (*Wheeler et al., 1994*; *Sugawara et al., 2007*), we found no cochlear dysfunction in mice lacking Ntf3 expression in these cells. The lack of phenotype in the hair cell-specific *Ntf3* knockout is unlikely due to the partial recombination in hair cells because (a) recombination in supporting cells and IHCs had similar efficiency (~60%); (b) cochlear Ntf3 expression was reduced to a similar extent by deletion from either supporting cells or hair cells; and (c) Ntf3 overexpression by hair cells had similar effects on cochlear function and synapse density as that seen in Ntf3 overexpression by supporting cells, indicating that modulation of Ntf3 expression in a subset of hair cells is sufficient to produce phenotypic outcomes. These observations support the conclusion that endogenous Ntf3 expressed by postnatal supporting cells, but not inner hair cells, is necessary for normal cochlear function.

The mosaic recombination pattern in the *Pou4f3/CreER*[T] Ntf3 overexpression revealed that the effects of Ntf3 on hair cell synaptogenesis are precisely localized, that is, synapse density was increased only on hair cells in which the transgene was activated. Thus, it appears that supporting cells create a physical barrier, allowing for local signaling events without cross talk between adjacent hair cells. The lack of effect of postnatal *Ntf3* knockout or overexpression on the total number of myelinated sensory axons in the osseous spiral lamina indicates that the changes in synapses is not due to alterations in the number of sensory neurons, and thus it is consistent with the notion that the effects of Ntf3 are

local. Present results suggest that Ntf3 levels may regulate the number of synapses by influencing branching of the unmyelinated terminals of cochlear sensory neurons.

Regardless of the cellular origin, the effects of *Ntf3* knockout or overexpression on synaptic density and cochlear function are restricted to the high-frequency (basal) half of the cochlea. The pro-survival effects of embryonic Ntf3 have the same tonotopy, that is, the loss of spiral ganglion cells in mice with *Ntf3* or *Ntrk3* knockout are primarily seen at the cochlear base (*Fritzsch et al., 1997*; *Tessarollo et al., 1997*; *Coppola et al., 2001*; *Farinas et al., 2001*). Thus, it appears that Ntf3 is the key endogenous trophic factor for both embryonic development and postnatal function of high-frequency cochlear neurons.

A previous study showed that Ntf3 promotes axonal growth and synaptogenesis in organ of Corti explants after excitotoxicity (*Wang and Green, 2011*). Our results indicate that in vivo, Ntf3 regulates ribbon synapse numbers without altering cochlear nerve axonal numbers, suggesting a direct effect of Ntf3 on IHC synaptogenesis. Most importantly, overexpression of Ntf3, but not Bdnf, promotes recovery from noise-induced synaptic degeneration and the associated decrements in auditory evoked potentials. Since the Ntf3 receptor Ntrk3 is expressed by cochlear neurons, not IHCs (*Gestwa et al., 1999*), our results suggest that Ntf3 overexpression acts first by promoting the recovery of post-synaptic terminals, which prevents the progressive loss of pre-synaptic ribbons, and enhances the regeneration of IHC synapses after noise exposure.

There has been extensive exploration of the use of Bdnf and Ntf3 as therapeutics for sensorineural hearing loss, based on their pro-survival effects on spiral ganglion neurons after hair cell degeneration due to ototoxic drugs (*Ramekers et al., 2012*). It has been suggested that these neurotrophins might promote long-term neuronal survival in cochlear implant users, who typically have few, if any, remaining hair cells (*Budenz et al., 2012*). Recent studies in mice have shown that significant synaptic loss precedes hair cell death and spiral ganglion cell degeneration in both noise-induced and age-related hearing loss (*Kujawa and Liberman, 2009*; *Sergeyenko et al., 2013*), and it has been suggested that this primary neuropathy is a major cause of problems hearing in a noisy environment, the most common complaint of those with sensorineural hearing loss. Therefore, treatment of cochlear synaptopathy presents a novel therapeutic approach for age-related and noise-induced hearing loss. While a number of trophic factors, including Bdnf and Ntf3, can mediate the survival of injured sensory neurons (*Roehm and Hansen, 2005*; *Ramekers et al., 2012*), our findings indicate that specific neurotrophins are required to promote regeneration and to regain function of the synaptic connections in damaged inner ear epithelia.

## Materials and methods

### Animals and tamoxifen treatments

*Plp1/CreER*[T] (*Doerflinger et al., 2003*), *Slc1a3/CreER*[T] (*Wang et al., 2012*), *Bdnf*[flox] (*Rios et al., 2001*), *Ntf3*[flox] (*Bates et al., 1999*), and *Rosa26*[tdTomato] reporter mice (*Madisen et al., 2010*) were obtained from Jackson Laboratory. The *Bdnf*[stop] mouse line was provided by Rudolf Jaenisch (*Chang et al., 2006*). To generate the *Ntf3*[stop] mouse, we engineered the conditional *Ntf3* overexpression transgene using the mouse *Ntf3* cDNA (kindly provided by Barbara Hempstead, Cornell University) under regulation by the synthetic CAGGS promoter/enhancer/intron followed by a loxP-STOP-loxP cassette (kindly provided by Laurie Jackson-Grusby, Children's Hospital Boston). Briefly, a loxP-STOP-loxP cassette was cloned immediately upstream of the *Ntf3* cDNA present in a Bluescript plasmid. A *SacI/KpnI* fragment containing the mouse *Ntf3* cDNA and the loxP-STOP-loxP cassette was cut out from the plasmid, blunt-ended, and ligated with an *EcoRI* cut/blunt-ended pCAGGSTurbo-cre vector to generate pCAGGS-loxP-STOP-loxP *Ntf3*. A *PstI* fragment containing an ATG-FRT site was cut out from the pPGK-ATG-FRT (no *EcoRI*) vector and subcloned into the *PstI* site, 3′ of the CAGGS-loxP-STOP-loxP *Ntf3* construct. The whole pCAGGS-loxP-STOP-loxP *Ntf3-ATG-FRT* plasmid was targeted downstream of the *collagen 1a1* locus by frt/Flpase-mediated site-specific integration. To generate *Pou4f3/CreER*[T] mice, an 8.6-kb *Pou4f3* (Brn3.1) regulatory sequence (*Sage et al., 2006*) was cut out from pSP73-*Pou4f3*-alpha9 plasmid using *SalI* (kindly provided by Doug Vetter, Tuffs University). The SP73-alpha9 fragment was re-ligated and digested with *SmaI* to remove alpha9 cDNA from SP73 backbone. The *CreER*[T] coding sequence (*Feil et al., 1996*) was obtained from *EcoRI* digestion of pCreERt plasmid (kindly provided by Laurie Jackson-Grusby), blunted and ligated to SP73 backbone. SP73-*CreER*[T] vector was then cut with *SalI* and re-ligated with *Pou4f3* regulatory region. The final plasmid (14.4-kb)

was digested with *FspI* and *SbfI* to release the 13-kb fragment containing *Pou4f3/CreER^T* transgene, which is then purified for microinjection. Both *Ntf3^stop* and *Pou4f3/CreER^T* mice were generated at Mouse Gene Manipulation Facility at Children's Hospital Boston.

To knockout *Bdnf* or *Ntf3* from postnatal supporting cells, we crossed *Bdnf^flox:Plp1/CreER^T* or *Ntf3^flox:Plp1/CreER^T* mice with *Bdnf^flox* or *Ntf3^flox* mice, respectively. Tamoxifen was injected intraperitoneally at 50 mg/kg/day from P0–P1 daily. To overexpress *Bdnf* or *Ntf3* from postnatal supporting cells, we crossed *Plp1/CreER^T* mice with *Bdnf^stop* or *Ntf3^stop* mice, respectively. Tamoxifen was injected at 33 mg/kg/day from P1–P7 daily. The recombination efficiency and specificity of *Pou4f3/CreER^T* mice were examined by crossing these mice with homozygous *Rosa26^tdTomato* reporter mice. To knockout *Ntf3* from hair cells, *Ntf3^flox:Pou4f3/CreER^T* mice were mated with *Ntf3^flox* mice. To overexpress *Ntf3* from hair cells, we crossed *Pou4f3/CreER^T:Rosa26^tdTomato* mice with *Ntf3^stop* mice. The hair cell-specific recombination was induced by tamoxifen injection at 50 mg/kg/day from P1–P3 daily. To overexpress *Ntf3* from adult supporting cells, we crossed *Slc1a3/CreER^T* mice with *Ntf3^stop* mice. *Ntf3^flox:Plp1/CreER^T* mice and their controls were on C57BL/6 background. *Ntf3^stop:Slc1a3/CreER^T* and their controls were on a mixed background of C57BL/6 and FVB/N. All the other mice were on FVB/N background.

For gene expression study, 10-week old mice were gavaged with tamoxifen (200 mg/kg/day) for 3 days. For acoustic trauma studies, 12-week old mice were gavaged with tamoxifen for 3 days with the first dosage given immediately (<1 hr) after noise exposure. In all experiments, *Cre* negative littermates were used as controls. All 'Materials and methods' were performed in compliance with animal protocols approved by the Institutional Animal Care and Use Committee at Children's Hospital Boston.

## RNA isolation and quantitative RT-PCR

Mice (8- to 10-week old) were euthanized in a $CO_2$ chamber and the inner ears were extracted. Cochleas (membranous labyrinths) and utricles (sensory epithelia) were dissected from temporal bones, and total RNA was purified using RNeasy spin-columns (Qiagen, Valencia, CA). Total RNA from cochlea (200 ng) or utricle (50 ng) was reverse transcribed (RT) using an iScript cDNA Synthesis Kit (Bio-Rad, Hercules, CA) in 20 µl reaction with a mixture of oligo(dT) and random hexamer primers. The reverse transcription was performed at 42°C for 1 hr followed by 85°C for 5 min. Quantitative PCR was carried out on a CFX96 machine using an iQ SYBR Green Supermix (Bio-Rad). For each well of the 96-well plate (Bio-Rad), the 20 µl reaction contained 10 µl of 2× iQ SYBR Green Supermix, 6 pmol of each forward and reverse primer, and 1 µl of cDNA sample. The cycling conditions were as follows: 95°C for 3 min followed by 40 cycles of 95°C for 30 s, 60°C for 30 s, and 72°C for 30 s. Each sample was loaded in duplicates. The following forward (F) and reverse (R) primers were used: *Bdnf*, F: GTGTGTGACAGTATTAGCGAGTGG, R: GATACCGGGACTTTCTCTAGGAC, which generates a 101 bp amplicon; *Ntf3*, F: GCCCCCTCCCTTATACCTAATG, R: CATAGCGTTTCCTCCGTGGT, which generate an 83 bp amplicon; and *Rpl19*, F: ACCTGGATGAGAAGGATGAG, R: ACCTTCAGGTACAGGCTGTG, which generates a 101 bp amplicon. Expression levels of Bdnf and Ntf3 were normalized to the reference housekeeping gene Rpl19 in the same samples. The relative expression level was calculated by the $2^{-\Delta\Delta Ct}$ method as shown previously (*Stankovic and Corfas, 2003*).

## Physiological analyses

Inner ear physiology, including vestibular evoked potentials (VsEPs, the summed activity of the vestibular afferent pathways to sudden head accelerations), auditory brainstem responses (ABRs, the summed activity of auditory afferent pathways to short tone bursts), and distortion product otoacoustic emissions (DPOAEs), was performed on mice anesthetized with xylazine (20 mg/kg, i.p.) and ketamine (100 mg/kg, i.p.). Analysis of animals without noise exposure was performed on 5- to 6-week old mice. For experiments involving noise exposure, the first recording was performed at 14–15 weeks of age, followed by noise exposure at 16 weeks of age and the additional measurements 1, 3, and 10 days post-exposure. For VsEPs and ABRs, needle electrodes were placed into the skin (a) at the dorsal midline close to the neural crest, (b) behind the left pinna, and (c) at the base of the tail (for a ground electrode). For VsEPs, mice were positioned on their backs, with the head coupled securely to a shaker platform. Stimuli were linear acceleration ramps, 2 ms in duration, applied in the earth–vertical axis at 17/s with alternating stimulus polarity. An accelerometer, mounted near the head, was used to calibrate the resultant jerk, which is expressed in dB *re* 1.0 g/ms. Electrophysiological activity was amplified (10,000×), filtered (0.3–3 kHz), and digitized (125 kHz), and 1024 responses were

averaged at each stimulus level. We collected an intensity series in 5 dB steps encompassing stimulus levels above and below threshold. ABR potentials were evoked with 5 ms tone pips (0.5 ms rise-fall, with a $cos^2$ envelope, at 40/s) delivered to the eardrum at log-spaced frequencies from 5.6 to 32 kHz. The response was amplified (10,000×) and filtered (0.3–3 kHz) with an analog-to-digital board in a PC-based data acquisition system. Sound level was raised in 5 dB steps from 10 to 80 dB sound pressure level (SPL). At each level, 1024 responses were averaged (with stimulus polarity alternated) after 'artifact rejection'. The DPOAEs in response to two primary tones of frequencies f1 and f2 were recorded at (2 × f1)–f2, with f2/f1 = 1.2, and the f2 level 10 dB is lower than the f1 level. The ear-canal sound pressure was amplified and digitally sampled at 4 μs intervals. DPOAE thresholds were defined as the f1 level required to produce a response at 0 dB SPL. These acoustic signals, generated by outer hair cells and measureable in the ear canal, are useful for differential diagnosis: attenuation of ABRs without a change in DPOAEs provides strong evidence for cochlear synaptic or neural dysfunction (*Kujawa and Liberman, 2009*).

## Noise exposure

Mice (at 12–16 weeks) were placed within small cells in a subdivided cage, suspended in a reverberant noise exposure chamber, and exposed to an octave band of noise (8–16 kHz) at 100 dB for 2 hr. Noise calibration to target SPL was performed immediately before each noise overexposure. Sound pressure levels varied by <1 dB across the cages.

## Cochlear morphology and axonal counts

Mice were perfused intracardially with 4% paraformaldehyde in 0.1 M phosphate buffer. Cochleas were extracted, perfused intralabyrinthly, and post-fixed with 1.5% paraformaldehyde and 2.5% glutaraldehyde. The cochleas were then osmicated in 1% osmium tetroxide, decalcified in 5% EDTA, dehydrated and embedded in araldite. Serial sections (20 μm) parallel to the modiolus were cut using a Leica RM2165 microtome and mounted on microscope slides in Permount. Cochlear regions of interest (6, 16, and 32 kHz) were identified based on 3D reconstruction and cochlear mapping (*Hirose and Liberman, 2003*). Axonal counts were made by imaging tangential sections through the osseous spiral lamina near the habenula perforata. Several fascicles of the cochlear nerve fibers were present in these sections. All myelinated fibers from each section were imaged using 63× DIC optics and counted. The number of axonal fibers was then divided by the cochlear length at specific frequency regions.

## Immunostaining, hair cell and synaptic counts

Analysis of animals without noise exposure was performed at 8–10 weeks. Analysis of noise-exposed animals was performed either immediately (AT + 2 hr) or 2 weeks after noise exposure (AT + 14 days). Cochleae were fixed as described above, post-fixed in 4% paraformaldehyde in 0.1 M phosphate buffer for 2 hr, and decalcified in 5% EDTA. Cochlear tissues were then microdissected and permeabilized by freeze-thawing in 30% sucrose. The microdissected pieces were blocked in 5% normal horse serum with 1% Triton X-100 in phosphate-buffered saline (PBS) for 1 hr, followed by incubation in primary antibodies (diluted in blocking buffer) at 37°C for 16 hr. The primary antibodies used in this study were: anti-myosin VIIa (rabbit anti-MyoVIIa; Proteus Biosciences, Ramona, CA; 1:500), anti-C-terminal binding protein 2 (mouse anti-CtBP2 IgG1; BD Biosciences, San Jose, CA; 1:200), and anti-glutamate receptor 2 (mouse anti-GluA2 IgG2a; Millipore, Billerica, MA; 1:2000). Tissues were then incubated with appropriate Alexa Fluor-conjugated fluorescent secondary antibodies (Invitrogen, Carlsbad, CA; 1:500 in blocking buffer) and 1 μg/ml DAPI (Invitrogen) for 1 hr at room temperature. The tissues were mounted on microscope slides in Vectashield mounting media (Vector laboratories, Burlingame, CA).

All pieces of each cochlea were imaged at low power to convert cochlear locations into frequency using a custom ImageJ plugin (http://www.masseyeandear.org/research/otolaryngology/investigators/laboratories/eaton-peabody-laboratories/epl-histology-resources/imagej-plugin-for-cochlear-frequency-mapping-in-whole-mounts). Confocal z-stacks of the 5.6, 8, 11.3, 16, 22.6, and 32 kHz regions from each cochlea were taken using a Zeiss LSM510 microscope equipped with either 40× (1× digital zoom, *Pou4f3/CreER^T* reporter study) or 63× (2× digital zoom, synaptic counts) oil immersion lens. The number of inner hair cells (IHCs) at specific cochlear regions was determined based on the DAPI nuclear counts at the IHC focal plane. For synaptic counts, the z-stacks (0.25 μm step size) were set to span the entire length of IHCs so that all the synaptic specializations were imaged. Image stacks were imported to Amira software (Visage Imaging, San Diego, CA), which produced three-dimensional (3D) renderings

of each confocal *z*-stack using the 'connected components' feature. CtBP2 and GluA2 puncta in each image stacks were then captured and counted automatically. To assess the appositions of CtBP2 with GluA2 puncta (putative synapses) or CtBP2/GluA2 puncta with tdTomato (synaptic puncta on recombined IHCs), the *z*-stacks were re-imaged using custom software (*Source Code 1*) that computed and displayed the *x-y* projection of the voxel space within 0.5 μm of the center of each puncta, as identified by Amira analysis (*Liberman et al., 2011*; *Lin et al., 2011*). The number of juxtaposed CtBP2 and GluA2 puncta or CtBP2/GluA2 puncta and tdTomato was visualized and counted from these miniature image arrays. Synaptic counts of each *z*-stack were divided by the number of IHC nuclei, which could be visualized by weak staining of CtBP2 antibody. Each image usually contained 8–10 IHCs.

## Acknowledgements

This research was supported in part by National Institute on Deafness and Other Communication Disorders Grants R01 DC 004820 (to GC) and P30 DC 005209 (to MCL), Eunice Kennedy Shriver National Institute of Child Health and Human Development Grant P30-HD 18655 (Mental Retardation Research Center) (to GC), the Department of Otolaryngology at Boston Children's Hospital and the Hearing Health Foundation (to GW).

## Additional information

### Funding

| Funder | Grant reference number | Author |
|---|---|---|
| National Institute on Deafness and Other Communication Disorders | R01 DC 004820 | Gabriel Corfas |
| National Institute of Child Health and Human Development | P30- HD 18655 (Mental Retardation Research Center) | Gabriel Corfas |
| Hearing Health Foundation | | Guoqiang Wan |
| National Institute on Deafness and Other Communication Disorders | P30 DC 005209 | M Charles Liberman |
| Boston Children's Hospital | Department of Otolaryngology | Guoqiang Wan |

The funders had no role in study design, data collection and interpretation, or the decision to submit the work for publication.

### Author contributions

GW, Acquisition of data, Analysis and interpretation of data, Drafting or revising the article; MEG-C, ARG, Acquisition of data, Analysis and interpretation of data; MCL, Conception and design, Drafting or revising the article; GC, Conception and design, Analysis and interpretation of data, Drafting or revising the article

### Ethics

Animal experimentation: This study was performed in strict accordance with the recommendations in the Guide for the Care and Use of Laboratory Animals of the National Institutes of Health. All of the animals were handled according to the approved institutional animal care and use committee (IACUC) protocol (#11-03-1911R) of Children's Hospital Boston.

### Author ORCIDs

Guoqiang Wan, http://orcid.org/0000-0003-3841-9301

## Additional files

### Supplementary file

• Source code 1. Amira and Blob Projection software.

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
