## [Decision Letter]

Congratulations: we are very pleased to inform you that your article, “Neurotrophin-3 regulates ribbon synapse density in the cochlea and induces synapse regeneration after acoustic trauma”, has been accepted for publication in *eLife*. The Reviewing Editor for your submission, Freda Miller, went over your rebuttal letter and response to the previous reviews, and reviewed the manuscript with the new data. She and the Senior editor felt that you had adequately addressed the only major technical comment from the previous reviews, which had to do with differences in baselines in the different genetic backgrounds. In addition, they both felt that the new data were very interesting. Based upon these new data, the previous excellence and rigor of the study as indicated by all three reviewers, and the thoughtful comments you had made concerning the differences between the current study and previously-published work, we felt that the manuscript now warranted publication in *eLife*.

[Editors’ note: this article was originally rejected after discussions between the reviewers, but the authors were invited to resubmit after an appeal against the decision.]

Thank you for choosing to send your work entitled “Neurotrophin-3 regulates ribbon synapse density in the cochlea and induces synapse regeneration after acoustic trauma” for consideration at *eLife.* Your full submission has been evaluated by a Senior editor and 3 peer reviewers, one of whom is a member of our Board of Reviewing Editors, and the decision was reached after discussions between the reviewers. We regret to inform you that your work will not be considered further for publication at this stage.

In particular, while all of the reviewers appreciated the elegant in vivo approach, and the high quality of the data, they also felt that in light of previously published work (detailed in the reviews), the current manuscript did not represent a substantive enough advance in the field for publication in *eLife.*

*Reviewer #1*:

In this manuscript by Wan et al., the authors have addressed a role for the neurotrophins NT-3 and BDNF in the postnatal ear by inducibly deleting or overexpressing NT-3 or BDNF in defined cell populations early in neonatal life. This is a very nicely-performed paper, and the data are largely convincing. Moreover, while there is already a lot known about BDNF and NT-3 during ear development, there are some interesting new conclusions here. The manuscript would, however, be strengthened by considering the following issues: 1) I am concerned about the control baselines in Figure 4-D, which show data that are very important to one of the main conclusions of the paper. In particular, the control NT-3flox is very different from the control NT-3stop, and indeed when the floxed NT-3 allele is inducibly deleted, then it looks just like the control NT-3stop. Which baseline is correct? While I realize there could be differences between different backgrounds, this is really a key issue. To clarify this and to make this important conclusion more definitive, it would be important for the authors to provide additional control populations. For example, they could characterize the PLP-creERT mice in the same genetic backgrounds, but without the NT-3flox or stop alleles.

2) The authors show nice data supporting the idea that NT-3 overexpression enhances synaptic regeneration after acoustic trauma. It would have added to the paper if the authors had asked whether NT-3 expressed by supporting cells was essential for synaptic regeneration by simply performing the same experiment in their NT-3flox mice.

3) It would be very helpful to prepare a table similar to Table 1 to directly compare the controls and NT-3 overexpressors statistically for the data shown in Figure 7.

*Reviewer #2*:

This is an elegant and well-designed study on the role of neurotrophin-3 (NT-3) and brain-derived neurotrophic factor (BDNF) in the postnatal inner ear. Using cell-specific and inducible knockout or overexpression technology, the authors demonstrate that NT-3 and BDNF expressed by supporting cells are required for the maintenance of ribbon synapses, a role distinct from that played during embryogenesis. Their findings show that these neurotrophins have segregated roles, with NT-3 having a major effect on cochlear function, while BDNF acts only in the vestibular organs. The central take-home message of this study is that NT-3 overexpression, but not BDNF, increases ribbon synapse density and elicits regeneration of synaptic contacts between hair cells and sensory neurons after acoustic trauma.

Recent studies indicate that synaptic loss precedes hair and spiral ganglion cell death in noise-induced and age-related hearing loss. Therefore, the study of signaling factors that regulate inner ear synapse density and synaptic regeneration after acoustic injury is highly significant. Overall, this work has numerous strengths including an extremely well written manuscript and robust data that include appropriate controls. The results are presented in carefully crafted, self-explanatory figures. The use of cell-specific inducible knockout or overexpressing mouse lines together with electrophysiological measurements in a model of acoustic trauma is very strong.

The overall approach of this study is more comprehensive and sophisticated than previous ones; however, a potential concern is the lack of novelty regarding the role of NT-3 on hair cell synapse regeneration. A previous report demonstrated that exogenous NT-3 increased synapse regeneration by spiral ganglion neurons onto inner hair cells after excitotoxic trauma in vitro (Wang and Green, 2011, J. Neurosci. 31:7938). Although there are several differences between the two studies, the primary finding is similar. For example, this previously published work showed that blocking endogenous NT-3 signaling with TrkC-IgG, but not TrkB-IgG, reduced post-synaptic density regeneration indicating that endogenous NT-3, but not BDNF, is required for synaptogenesis. These data are in agreement with the work presented here, but novelty (or lack of) might be an issue.

The use of PSD95 as a post-synaptic marker is more common than GluA2 and might be more appropriate to compare current data with previous work.

*Reviewer #3*:

The authors have used a variety of cell-type specific inducible knockout or overexpressor mice to examine the roles of BDNF and NT-3 produced by hair cells or supporting cells in the postnatal inner ear. They show convincingly that supporting cell-derived BDNF and NT3 are required for the maintenance of ribbon synapses in the vestibular organs and cochlea, respectively, without affecting neuron or axon number. They show that the effects of NT3 on cochlear function and synaptic density and are restricted to the high-frequency, basal region of the cochlea. Furthermore, they show that NT3 overexpression is effective in promoting recovery from noise-induced synaptic degeneration, raising the possibility of developing a novel therapeutic approach for noise-induced hearing loss based on these findings.

The paper is a technical tour de force, the data are clearly illustrated and the paper is concise and well-written. The findings significantly and usefully extend our understanding of the specific and selective functions of neurotrophins in the mature inner ear, and as such represent an important contribution to the field with potential clinical relevance. I have no major criticisms of this work.

---

## [Author Response]

Reviewer #1:

*[…] 1) I am concerned about the control baselines in*
Figure 4*-D, which show data that are very important to one of the main conclusions of the paper. In particular, the control NT-3flox is very different from the control NT-3stop, and indeed when the floxed NT-3 allele is inducibly deleted, then it looks just like the control NT-3stop. Which baseline is correct? While I realize there could be differences between different backgrounds, this is really a key issue. To clarify this and to make this important conclusion more definitive, it would be important for the authors to provide additional control populations. For example, they could characterize the PLP-creERT mice in the same genetic backgrounds, but without the NT-3flox or stop alleles*.

The reviewer raised a valid point. We have now compared the synapse density in the background strains for both NT -3flox (C57BL/6J) and NT-3stop (FVB/N) lines (see Figure 9) and found that C57BL/6J cochleae have higher synaptic density than FVB/N. This explains the different synapse baselines observed between NT-3flox and NT-3stop mice (Figure 4 D). We now state in the Materials and methods section the specific genetic background for each group of mice.Author response image 1.Ribbon synaptic density of C57BL/6J and FVB/N mouse strains. (**A**) Representative confocal images (maximal projection from a focal series) of IHC synapses from 32 kHz region of 8 wk-old C57BL/6J and FVB/N cochleae immunolabeled for pre-synaptic ribbons (CtBP2-red) and post-synaptic receptor patches (GluA2-green) (scale bar = 10 µm). (**B**) Synaptic counts of C57BL/6J and FVB/N cochleae at 32 kHz; *n* = 4-6. *p<0.05 by unpaired student’s *t*-test.

*2) The authors show nice data supporting the idea that NT-3 overexpression enhances synaptic regeneration after acoustic trauma. It would have added to the paper if the authors had asked whether NT-3 expressed by supporting cells was essential for synaptic regeneration by simply performing the same experiment in their NT-3flox mice*.

In the second part of the study (acoustic trauma), our primary aim is to address the clinical potential of NT-3 (and BDNF) in promoting synaptic regeneration and hearing restoration after injury. While it is an interesting question to ask if the endogenous level of NT-3 can affect the spontaneous recovery of synapses, we felt it is not a crucial experiment for extending clinical application of NT-3.

*3) It would be very helpful to prepare a table similar to*
Table 1
*to directly compare the controls and NT-3 overexpressors statistically for the data shown in*
Figure 7.

We have now generated a table (Table 2) to present the statistical difference between controls and NT-3 overexpressors for Figure 7.

Reviewer #2:

*[…] The overall approach of this study is more comprehensive and sophisticated than previous ones; however, a potential concern is the lack of novelty regarding the role of NT-3 on hair cell synapse regeneration. A previous report demonstrated that exogenous NT-3 increased synapse regeneration by spiral ganglion neurons onto inner hair cells after excitotoxic trauma* in vitro *(Wang and Green, 2011, J. Neurosci. 31:7938). Although there are several differences between the two studies, the primary finding is similar. For example, this previously published work showed that blocking endogenous NT-3 signaling with TrkC-IgG, but not TrkB-IgG, reduced post-synaptic density regeneration indicating that endogenous NT-3, but not BDNF, is required for synaptogenesis. These data are in agreement with the work presented here, but novelty (or lack of) might be an issue*.

It is true that exogenous NT-3 has been shown to promote afferent reinnervation and synaptic regeneration in cochlear explant cultures. However, we believe that the significance and implications of our study, which focus on the adult and exposure to noise, are vastly different from the Wang and Green paper.

a) The Wang and Green paper is based on neonatal tissues, a time at which cochlear synapses are immature and even hair cell regeneration occurs spontaneously. Therefore, the conclusions of that paper are relevant to synapse development, not necessarily for synapse regeneration in the adult inner ear.

b) We show for the first time that the endogenous NT-3 from supporting cells surrounding the IHCs are important for formation/maintenance of the ribbon synapses.

c) By working in intact animals, we demonstrate that functional relevance of NT-3 on synapse formation and regeneration in the context of hearing in the normal and injured inner ear.

d) While tissue culture studies can provide important insights into biological process and mechanisms, there are many examples of conclusions reached by in vitro studies that were eventually invalidated by in vivo experiments. We believe that showing that NT-3 can alter the consequences of noise exposure in the adult is novel and has clinical implications for the development of treatments for sensorineural hearing loss that go beyond what the Wang and Green paper provided.

*The use of PSD95 as a post-synaptic marker is more common than GluA2 and might be more appropriate to compare current data with previous work*.

We have previously shown that pre-synaptic ribbons (labeled by CtBP2) of inner hair cell-spiral ganglion neuron synapses have similar juxtaposition to PSD95 (Yuan Y et al. JARO 2013) and GluA2 (Liberman *LD* et al. J Neurosci. 2011) We think that data from this study, specifically the ribbon synapses labeled by CtBP2/GluA2 can be readily compared to previous work using PSD95 instead of GluA2.